# LaCache: Ladder-Shaped KV Caching for Efficient Long-Context Modeling of Large Language Models

Dachuan Shi [1]   Yonggan Fu [1 2]   Xiangchi Yuan [1]   Zhongzhi Yu [1]   Haoran You [1]   Sixu Li [1]   Xin Dong [2]   Jan Kautz [2]
Pavlo Molchanov [2]   Yingyan (Celine) Lin [1 2]

## Abstract

Recent advancements in Large Language Models (LLMs) have spurred interest in numerous applications requiring robust long-range capabilities, essential for processing extensive input contexts and continuously generating extended outputs. As sequence lengths increase, the number of Key-Value (KV) pairs in LLMs escalates, creating a significant efficiency bottleneck. In this paper, we propose a new KV cache optimization paradigm called **LaCache**, a training-free method for efficient and accurate generative inference of LLMs. LaCache enables LLMs to simultaneously address both of the critical challenges in long-range modeling: robust long-range capabilities and continuous generation without running out-of-memory (OOM). Specifically, LaCache integrates two key innovations: (1) a ladder-shaped KV cache pattern that stores KV pairs not only sequentially (left-to-right within each layer) but also across layers (from shallow to deep), providing an extended span for capturing long-range dependencies under a fixed storage budget, thereby boosting long-range capabilities; and (2) an iterative compaction mechanism that progressively compresses older caches, freeing up space for new tokens within a fixed cache size. This token distance-based dynamic compression enables more effective continuous generation under constrained cache budgets. Experiments across various tasks, benchmarks, and LLM models consistently validate LaCache's effectiveness in enhancing LLMs' long-range capabilities. Our code is available at https://github.com/GATECH-EIC/LaCache.

[1]Georgia Tech [2]NVIDIA. Correspondence to: Yingyan (Celine) Lin <celine.lin@gatech.edu>.

*Proceedings of the 42ⁿᵈ International Conference on Machine Learning*, Vancouver, Canada. PMLR 267, 2025. Copyright 2025 by the author(s).

## 1. Introduction

Large Language Models (LLMs) have significantly advanced natural language processing tasks (Achiam et al., 2023; Touvron et al., 2023; Jiang et al., 2023; Hurst et al., 2024; Team et al., 2024; OpenAI, 2025; Guo et al., 2025; Gemini, 2025; Anthropic, 2025), but they also face major challenges due to their substantial computational costs. To mitigate this, key-value (KV) caching has been used to avoid recomputing attention keys and values during the auto-regressive decoding of LLMs. However, this approach introduces significant memory overhead that scales linearly with sequence length, leading to out-of-memory (OOM) issues on long sequences.

Existing KV cache eviction strategies attempt to address these challenges by pruning cached KV states to enhance memory efficiency. However, these strategies often struggle to balance two critical requirements for long-range LLMs in real-world applications: robust long-range capabilities and continuous generation without OOM. For example, StreamingLLM (Xiao et al., 2023b) prioritizes continuous generation but compromises accuracy on long-context tasks. Quest (Tang et al., 2024) maintains high accuracy but at the cost of substantial memory usage due to the need to cache the entire KV cache, eventually leading to OOM on long sequences. H2O (Zhang et al., 2024) reduces memory costs and achieves better accuracy than StreamingLLM, but its reliance on attention maps makes it incompatible with the efficient attention implementation FlashAttention (Dao, 2023), leading to slow attention computation.

In response to the aforementioned limitations, we propose LaCache, a training-free KV cache optimization framework that employs a ladder-shaped storage pattern for accurate and efficient generative inference in LLMs. Our contributions are summarized as follows:

- We propose LaCache, which introduces a novel ladder-shaped KV cache pattern designed for accurate and cost-effective long-context generation. This strategy stores KV pairs not only sequentially (left to right within each layer) but also across layers (from shallow to deep). This configuration extends the span for capturing long-range

dependencies under a constrained storage budget, thereby enhancing long-range capabilities. Specifically, it preserves the KV states of early tokens in earlier layers and progressively shifts the focus to later tokens in subsequent layers, forming a stepwise, ladder-like structure. As analyzed later, the ladder-shaped pattern improves the lower bound of overall information retention across all tokens.

- We further integrate LaCache with an iterative compaction mechanism to support continuous generation for infinitely long sequences without OOM. This approach periodically applies a ladder-based compression pattern to previously condensed KV states, freeing up space for new tokens. It compresses older tokens more aggressively while applying less compression to newer tokens, enabling the model to prioritize recent information while efficiently managing memory for incoming tokens.

- We evaluate and validate the effectiveness of LaCache through a series of experiments and ablation studies. Results across multiple benchmarks consistently demonstrate that LaCache enhances long-range capabilities and supports continuous generation. Additionally, due to its compatibility with FlashAttention, it outperforms importance-based methods such as H2O in terms of achievable accuracy-throughput trade-offs.

## 2. Related Work

**Long-context LLM.** The demand for long-context modeling has surged due to its ability to handle complex, multi-step tasks and maintain coherent interactions. This has spurred extensive research on enhancing long-context generation (Li et al., 2023; Peng et al., 2023; Ye et al., 2025), enabling models to process more tokens per forward pass. While approximate attention mechanisms (Beltagy et al., 2020; Kitaev et al., 2020; Wang et al., 2020) have improved efficiency, they usually lead to degradation in task accuracy. Recent advances in positional embeddings, such as position interpolation and fine-tuning (Chen et al., 2023; Peng & Quesnelle, 2023), have further extended context windows. However, inference efficiency remains a bottleneck for long input sequences. In our proposed LaCache, we leverage token eviction to enhance efficiency in long-context generation, including continuous or infinite generation tasks.

**KV cache eviction.** KV cache eviction techniques mitigate excessive cache growth by removing non-essential tokens. Early methods (Xiao et al., 2023b; Han et al., 2024) rely on static, naive retention strategies that overlook model processing patterns and input context, leading to accuracy degradation. To improve long-context handling, dynamic approaches (Adnan et al., 2024; Liu et al., 2024; Zhang et al., 2024; Wan et al., 2024) utilize attention weights to identify important tokens. For example, Liu et al. (2024) detects repetitive patterns to estimate token significance, while Wang et al. (2024); Shi et al. (2024); Yu et al. (2024) merge similar tokens. However, these methods depend on full attention weights, making them incompatible with state-of-the-art (SOTA) efficient inference frameworks such as FlashAttention (Dao et al., 2022), which do not explicitly compute attention maps. This constraint limits their real-device efficiency. To overcome this, LaCache introduces an attention-free KV cache eviction strategy, ensuring high accuracy while maintaining real-device efficiency.

**Efficient LLM inference.** While traditional efficiency techniques such as quantization, pruning, and distillation (Frantar et al., 2022; Lin et al., 2024; Fu et al., 2024b; Shi et al., 2023; Zhang et al., 2023; Yuan et al., 2025) remain valuable, system-level optimizations, such as FlashAttention (Dao et al., 2022; Dao, 2023; Shah et al., 2024) and memory offloading (Sheng et al., 2023), have emerged as key enablers for large efficiency gains in LLM inference. Unlike most dynamic KV cache eviction methods, LaCache can be seamlessly integrated with these system-level approaches, enhancing real-device efficiency with SOTA inference frameworks.

## 3. The Proposed LaCache Framework

In this section, we present the proposed LaCache framework. We begin with an overview of LaCache in Sec. 3.1, followed by an introduction to the proposed ladder-shaped KV cache patterns in Sec. 3.2. Finally, to support efficient continuous generation without OOM, even for tasks involving infinite-length generation, we further enhance LaCache with an iterative compaction strategy, as described in Sec. 3.3.

### 3.1. LaCache: Motivation and Methodology Overview

**Limitations of existing methods.** Existing efficient generation methods for LLMs face limitations in achieving both generation accuracy and memory efficiency, which are crucial for continuous long-context generation without running out of memory. Specifically, as shown in Fig. 1 (a), recency-based methods like StreamingLLM (Xiao et al., 2023b), which only maintain the KV cache of the latest tokens within a fixed-length sliding window with an $\mathcal{O}(1)$ memory complexity, can support infinite-length generation without OOM but may compromise generation accuracy. In contrast, retrieval-based methods like Quest (Tang et al., 2024), as shown in Fig. 1 (b), store the full KV cache of all tokens and retrieve the most relevant ones for each new token on the fly to improve computational efficiency. This strategy can achieve high accuracy across tasks due to the maintenance of the entire KV cache, but suffers from massive cache storage with an $\mathcal{O}(T)$ memory complexity, leading to potential OOM issues when handling long contexts.

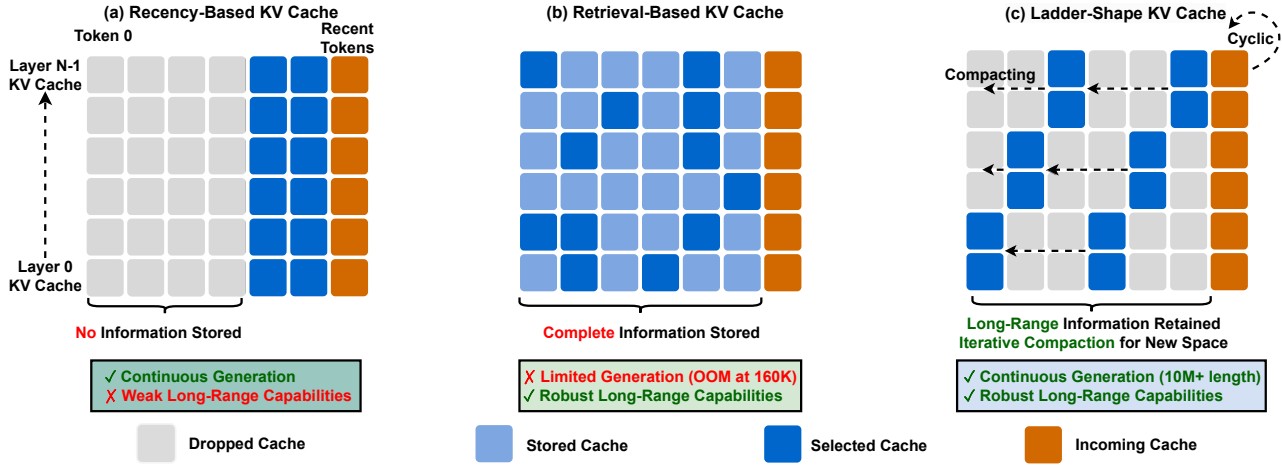

Figure 1. Illustrative comparisons among (a) recency-based KV cache (Xiao et al., 2023b), (b) retrieval-based KV cache (Tang et al., 2024), and (c) our proposed LaCache featuring a ladder-shaped pattern. Previous KV cache storage strategies struggle to simultaneously balance the needs for both continuous generation without OOM and robust long-range capabilities. In contrast, our proposed LaCache allows LLMs to simultaneously satisfy the two requirements.

In light of the limitations of both approaches, as shown in Fig. 1 (c), we propose LaCache, a training-free KV cache optimization featuring a ladder-shaped pattern, designed to balance accuracy and storage cost, enabling accurate and continuous generation without suffering from OOM.

**Overview.** Motivated by the need for both accurate and efficient KV caching, our proposed LaCache features a ladder-shaped KV cache compression and storage pattern. We illustrate LaCache's pattern in Sec. 3.2. Specifically, to achieve effective KV compression while preserving important information of past tokens, rather than uniformly keeping the KV cache for the same set of tokens across all layers as in StreamingLLM (Xiao et al., 2023b), we preserve the KV states of early tokens in earlier layers and then progressively shifting the focus to later tokens in the subsequent layers, forming a stepwise, ladder-like structure.

Furthermore, to support continuous generation without suffering from OOM even for infinite-length generation, we augment LaCache with an iterative compaction strategy, as depicted in Sec. 3.3. Specifically, each time the KV cache reaches its capacity, we apply our LaCache with the ladder-shaped pattern to the already-compacted KV cache. This strategy ensures that older token information is progressively compressed further, while newer incoming tokens are compressed less. In the following subsections, we will elaborate on the ladder-shaped pattern and the iterative compaction strategy, which are the two key enablers of our LaCache framework.

### 3.2. LaCache: Ladder-Shaped KV Cache Pattern

**The key insight.** Unlike StreamingLLM (Xiao et al., 2023b), which maintains the KV cache of the same set

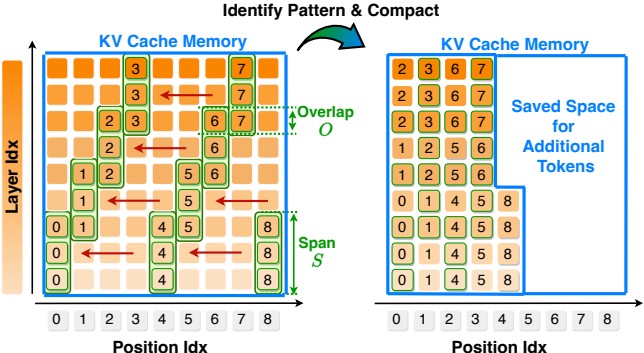

Figure 2. An illustration of LaCache's KV cache storage pattern. LaCache is leveraged to compact the original full KV cache into a compressed, ladder-shaped pattern, allowing for the storage of information from longer-range tokens compared to StreamingLLM (Xiao et al., 2023b) under the same KV cache budget, thereby providing stronger long-range sequence modeling.

of most recent tokens across all layers, our key insight is that while the information from recent tokens is critical for generation accuracy, their KV cache can be maintained and processed by fewer layers. In other words, different layers can maintain the KV cache corresponding to different sets of tokens. The key advantage of this approach is that, under the same KV cache budget, more tokens can be retained in the KV cache, effectively enlarging the context length and preserving more past information.

**The proposed ladder-shaped pattern.** The aforementioned insight inspires the design of our ladder-shaped KV cache pattern. As shown in Fig. 1 (c), our LaCache adopts a simple yet effective strategy to cache the KV states of varied tokens across different layers: it preserves the KV states of early tokens in earlier layers and then progressively shifts

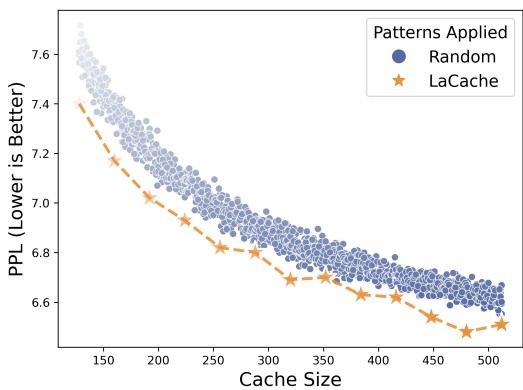

*Figure 3.* Visualize the trade-off between PPL and cache size for LaCache and over 1,500 randomly sampled KV cache patterns.

focus to later tokens in the subsequent layers, aligning with the temporal and sequential processing nature. This approach results in our ladder-shaped pattern, ensuring both storage efficiency and the retention of essential information across past tokens. More specifically, as shown in Fig. 2, to implement the ladder-shaped pattern denoted by the green box, we discard the KV states falling outside[1] this pattern and then condense the original 2D KV cache into a more compact structure with reduced cache size.

**Further analysis.** We note that the ladder-shaped pattern can effectively cover potentially important tokens by improving the lower bound of information retention. Our work intentionally does not rely on attention maps to identify important tokens, thereby avoiding conflicts with existing optimizations designed for efficient attention calculation. Two rationales behind our ladder-shaped KV pattern are:

Firstly, continuously extending a repetitive pattern and assigning coverage as equally as possible to each layer, which is ensured by our ladder-shaped pattern, improves the lower bound of information retention. This is because, in the worst case, important token sets may appear in the layer with the least coverage, and an unequal coverage strategy would lead to an accuracy drop.

Secondly, since neighboring tokens in natural language typically have higher semantic relevance, our ladder-shaped pattern incorporates a smooth transition for each preserved cache segment. As a result, the ever-expanding ladder pattern with partial overlaps enables a smoother fade-out of older tokens, maintaining stable information retention.

To empirically verify the benefits of these rationales, we randomly generate over 1500 patterns under different KV cache sizes to explore various configurations and visualize the achieved trade-off between Perplexity (PPL) and cache size in Fig. 3. As shown, our ladder-shaped pattern lies on the Pareto optimality boundary.

---

[1] To avoid bubbles, slightly more positions are preserved for tokens located at the beginning and end of ladders.

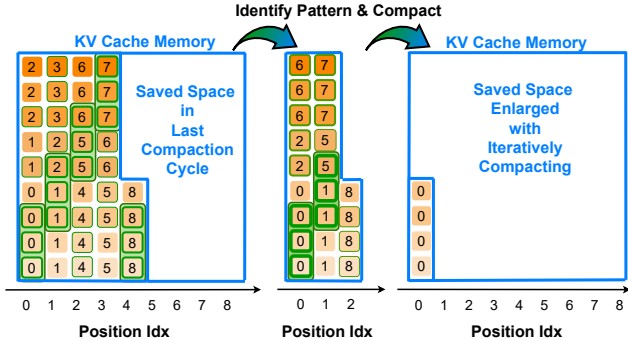

*Figure 4.* An illustration of LaCache's iterative compaction. Iterative compaction is introduced to support continuous generation without running out of memory, even for infinite-length generation. Once the KV cache reaches the predefined size, LaCache's ladder-shaped pattern is applied to the already-compacted KV cache, freeing up space for new tokens.

**Implementation details.** To balance storage efficiency and generation accuracy, we need to ensure that our method properly eliminates redundancy in the stored KV states while accurately preserving past context information by retaining sufficient past KV states.

To satisfy these principles, two key design factors help balance both aspects and achieve an optimal trade-off: (1) the *span S* across consecutive layers, *i.e.*, the number of layers used to preserve the KV state corresponding to the same token, and (2) the *overlap O* of each layer, *i.e.*, the number of tokens with their KV states preserved in each layer. The larger the *span S*, the more layers are used to record the context for the same token, thereby improving context preservation with an increased storage cost. Similarly, the larger the *overlap O*, the more KV states are preserved by each layer, allowing for more accurate context recording at the cost of reduced storage efficiency. As demonstrated in Sec. 4, we calibrate these two design factors to minimize cache redundancy and maximize generation accuracy.

### 3.3. LaCache: Iterative Compaction for Continuous Infinite-Length Generation

To enable continuous generation in LLMs without encountering OOM issues, even with an infinite generation length, it is highly desirable to maintain a constant KV cache size. To achieve this, we need to augment our LaCache with an eviction mechanism that removes the KV states of past tokens when the predefined cache size is fully utilized.

**Rationale and advantage of our method.** In response to the aforementioned need, we propose an iterative compaction strategy. The rationale behind our method is simple: once the KV cache, already condensed using LaCache, is full, we apply LaCache again to further condense it.

The advantages of this approach include: (1) Thanks to La-

*Table 1.* Language modeling experiments are conducted on the concatenated Wikitext-2-raw-v1 dataset. We use decoding lengths ranging from 1K to 16K for each tested model and cache budget. When the decoding length exceeds the pre-training length, models using a full cache encounter a perplexity explosion issue. The numbers in brackets indicate the cache size.

| Model | Decoding Length | | | | | Model | Decoding Length | | | | |
|---|---|---|---|---|---|---|---|---|---|---|---|
| | 1K | 2K | 4K | 8K | 16K | | 1K | 2K | 4K | 8K | 16K |
| Llama2-7B (100%) | 4.02 | 4.18 | 5.12 | nan | nan | Llama2-7B-Chat (100%) | 4.94 | 5.32 | 6.52 | nan | nan |
| *w/* StreamingLLM (512) | 5.54 | 5.84 | 6.32 | 6.93 | 5.36 | *w/* StreamingLLM (512) | 6.67 | 7.41 | 7.95 | 8.97 | 6.98 |
| *w/* LaCache (512) | **4.53** | **5.00** | **5.81** | **6.61** | **5.19** | *w/* LaCache (512) | **5.20** | **6.01** | **7.06** | **8.35** | **6.64** |
| *w/* StreamingLLM (256) | 6.08 | 6.38 | 6.90 | 7.52 | 5.92 | *w/* StreamingLLM (256) | 7.68 | 8.45 | 8.98 | 9.89 | 7.88 |
| *w/* LaCache (256) | **5.57** | **5.98** | **6.60** | **7.34** | **5.77** | *w/* LaCache (256) | **7.16** | **7.91** | **8.47** | **9.57** | **7.53** |
| Llama2-13B (100%) | 3.89 | 3.70 | 4.67 | 134.25 | nan | Llama3-8B (100%) | 4.28 | 4.39 | 5.82 | 6.16 | 109.94 |
| *w/* StreamingLLM (512) | 4.92 | 5.02 | 5.66 | 6.28 | 4.82 | *w/* StreamingLLM (512) | 5.46 | 5.33 | 6.73 | 6.99 | 5.52 |
| *w/* LaCache (512) | **4.40** | **4.68** | **5.42** | **6.09** | **4.69** | *w/* LaCache (512) | **4.61** | **4.89** | **6.40** | **6.78** | **5.40** |
| *w/* StreamingLLM (256) | 5.64 | 5.52 | 6.17 | 6.79 | 5.29 | *w/* StreamingLLM (256) | 6.39 | 5.97 | 7.40 | 7.66 | 6.06 |
| *w/* LaCache (256) | **5.22** | **5.17** | **5.88** | **6.50** | **5.08** | *w/* LaCache (256) | **5.71** | **5.61** | **7.11** | **7.38** | **5.86** |

Cache's ladder-shaped pattern design introduced in Sec. 3.2, early KV caches are discarded first when applying LaCache on the already-condensed KV cache, as shown in Fig. 4. This use of larger/smaller compression ratios on early/late KV caches aligns with the design principles of recency-based methods (Xiao et al., 2023b); (2) From a deployment perspective, iterative compaction using LaCache provides a unified solution and a clean interface, facilitating wider use.

**Implementation details.** We provide a more detailed illustration of how our iterative compaction works. As shown in Fig. 4, parts of the stored KV states are highlighted to demonstrate their changes after iterative compaction, while the non-highlighted parts are also occupied by other KV states. When the KV cache reaches its capacity, LaCache is applied to the stored KV states, which have already been condensed by LaCache when first enqueued into the KV cache. Consequently, KV states that fall outside the ladder-shaped pattern are discarded, and the freed space is allocated for new incoming tokens, thus enabling continuous infinite-length generation. In the second iteration of Fig. 4, older KV states are compressed more while newer ones are compressed less, thus better preserving recent information.

## 4. Experimental Results

### 4.1. Experimental Settings

**Models.** To validate LaCache's general effectiveness, we apply it to LLMs with varying sizes and functionalities, including Llama2-7B/13B (Touvron et al., 2023), Llama3-8B (Dubey et al., 2024), Llama2-7B/13B-Chat (Touvron et al., 2023), Llama3.2-3B-Instruct (Dubey et al., 2024), SmolLM2-1.7B-Instruct (Allal et al., 2024), and LongChat-7b-v1.5 (Li et al., 2023).

**Datasets.** We evaluate LaCache on two tasks: long-context

modeling and long-context understanding. Specifically, for long-context modeling, we use Wikitext-2 (Merity, 2016) and PG19 (Rae et al., 2019) datasets for evaluation. For long-context understanding, we employ LongBench (Bai et al., 2023), Needle-In-A-Haystack (Fu et al., 2024a), and RULER (Hsieh et al., 2024) benchmarks to thoroughly assess LaCache's achievable performance.

**Baselines.** We benchmark LaCache against the standard full KV cache settings and prior KV cache compression methods, including StreamingLLM (Xiao et al., 2023a), H2O (Zhang et al., 2024), TOVA (Oren et al., 2024), PyramidInfer (Yang et al., 2024), and SnapKV (Li et al., 2024) under various cache budgets.

**Implementation Details.** We implement LaCache in PyTorch (Paszke, 2019). For all tasks, we use a batch size of 1 for evaluation. Specifically, for language modeling experiments, we follow the implementation in StreamingLLM (Xiao et al., 2023b) and H2O (Zhang et al., 2024), employing regular token-by-token generation for Wikitext-2 and a sliding window approach with a window length of 256 tokens for PG-19. For all 21 datasets within the LongBench benchmark, following LongBench's default setting, we retain the first 128 tokens unchanged for both LaCache and baseline methods, as these initial tokens primarily consist of system prompts and questions. For the Needle-In-A-Haystack (Fu et al., 2024a) benchmark, we adopt 50 repetitions for each test unit and a context length up to 128k. For the RULER (Hsieh et al., 2024) benchmark, we adopt 100 repetitions for each test unit and a context length of 16k.

### 4.2. Long-Context Modeling Benchmarks

**Benchmark on Wikitext-2.** We first evaluate LaCache on the concatenated Wikitext-2-raw-v1 dataset in a standard

token-by-token generation setting. Four models, Llama2-7B, Llama2-7B-Chat, Llama2-13B, and Llama3-8B, are tested with KV cache budgets of 256 and 512.

As summarized in Tab. 1, our experimental results demonstrate that our LaCache consistently shows stronger capabilities in capturing long-range dependencies compared to the recency-based method StreamingLLM (Xiao et al., 2023b), under the same KV cache budget, across decoding lengths ranging from 1K to 16K. Specifically, with a KV cache budget of 512 and a 1K-length input, LaCache only degrades perplexity by $(5.20 - 4.94)/4.94 \approx 5\%$ on Llama2-7B-Chat compared to using the full cache. In contrast, StreamingLLM (Xiao et al., 2023b), under the same cache budget, results in a $(6.67 - 4.94)/4.94 \approx 35\%$ degradation in perplexity. This indicates that when targeting $2\times$ KV cache compression, LaCache experiences significantly less degradation in language modeling performance (5% vs. 35%) compared to StreamingLLM (Xiao et al., 2023b).

**Benchmark on PG19.** To assess LaCache's capabilities on language modeling tasks with extremely long inputs, we compare it against full cache and StreamingLLM on the concatenated PG19 dataset, which comprises 100 books totaling 10 million tokens. We adopt a sliding window of 256 tokens for higher efficiency, following the settings in (Wolf, 2019). The compacted KV cache output from each window is then passed to generate subsequent tokens, allowing the evaluation of the model's long-context capabilities.

As shown in Fig. 5, after an 8K input length, the perplexity of the Llama3-8B model using a full cache quickly escalates; after a 160K input length, an OOM issue arises on a single NVIDIA A100 GPU. In contrast, with our LaCache, the Llama3-8B model supports continuous generation with up to a 600K input length while maintaining reasonable perplexity.

Furthermore, Fig. 6 summarizes the comparison between LaCache and StreamingLLM (Xiao et al., 2023b) on the fully concatenated PG19 dataset. The consistent improvements achieved by LaCache further validate its strong long-range capabilities with inputs exceeding 10 million tokens.

**Benchmark under an extremely small cache budget.** To demonstrate LaCache's effectiveness under an extremely small cache budget, we further apply it to LLaMA3-8B-8K with a cache budget of 80. As shown in Tab. 2, LaCache

*Table 2.* Evaluate LaCache on the Llama3-8B-8K model using a cache size equal to 1% of the pre-training sequence length (*i.e.*, 80 tokens).

| Decoding Length | 1K | 2K | 4K | 8K | 16K | 32K | 64K | 128K |
|---|---|---|---|---|---|---|---|---|
| Llama3-8B | | 4.28 | 4.39 | 5.82 | 6.16 | 109.94 | nan | nan | nan |
| *w/* StreamingLLM | 7.28 | 7.78 | 8.31 | 8.73 | 8.88 | 8.88 | 9.94 | 15.68 |
| *w/* LaCache | **7.13** | **7.44** | **7.99** | **8.36** | **8.46** | **8.43** | **9.53** | **15.08** |

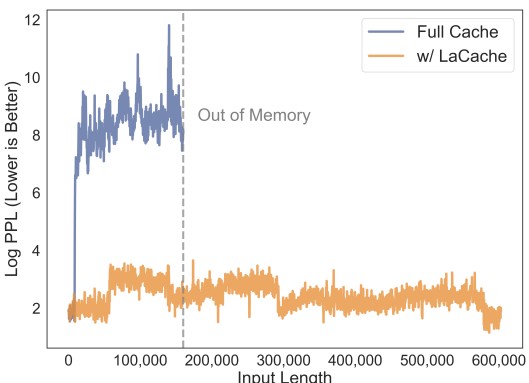

*Figure 5.* Evaluate LaCache on the first ten books of the concatenated PG19 dataset, corresponding to a length of 600K tokens.

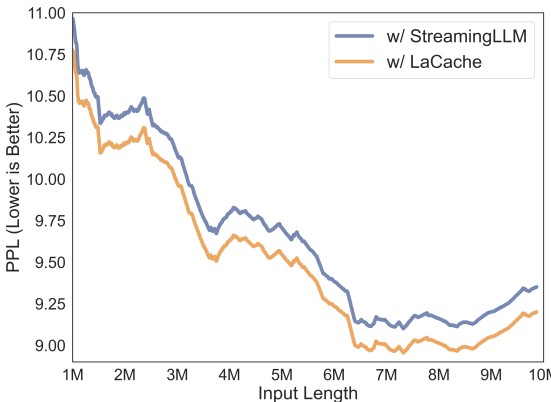

*Figure 6.* Evaluate LaCache on the entire concatenated PG19 dataset, corresponding to a length of 10 million tokens.

consistently achieves lower PPL than StreamingLLM and the full cache setting under the same decoding length.

### 4.3. Long-Context Understanding Benchmarks

**Benchmark on LongBench.** The LongBench benchmark (Bai et al., 2023) evaluates LLMs' bilingual long-context understanding capabilities, with most task lengths averaging between 5K and 15K tokens. Evaluation results on the 21 LongBench datasets for the LLaMA2-7B/13B-Chat models are presented in Tab. 3, and results for the SmolLM2-1.7B-Instruct model are shown in Tab. 4.

Our experimental results demonstrate that LaCache, requiring no additional computation or storage costs, outperforms StreamingLLM under the same KV cache budgets. For instance, with a 50% KV cache budget, LaCache reduces the average performance degradation from StreamingLLM's 2.4 to 1.5 on the LLaMA2-13B-Chat model, from 2.5 to 1.7 on the LLaMA2-7B-Chat model, and from 1.5 to 1.0 on the SmolLM2-1.7B-Instruct model.

**Benchmark with more KV cache eviction methods.** To further compare with attention-based KV cache eviction methods, we evaluate the trade-offs between task perfor-

*Table 3.* Evaluate LaCache on Llama2-7B/13B-Chat models across 21 LongBench datasets under 50% and 25% KV cache budgets.

| Model | Llama2-7B-Chat | | | | | Llama2-13B-Chat | | | | |
|---|---|---|---|---|---|---|---|---|---|---|
| | 100% | StreamingLLM | | LaCache | | 100% | StreamingLLM | | LaCache | |
| Cache budget | | 50% | 25% | 50% | 25% | | 50% | 25% | 50% | 25% |
| HotpotQA | 33.84 | 29.98 | 30.74 | 32.62 | 30.60 | 38.86 | 37.09 | 37.16 | 38.08 | 36.49 |
| 2WikiMultihopQA | 26.83 | 24.75 | 24.99 | 26.22 | 25.19 | 34.19 | 32.30 | 32.15 | 34.11 | 31.17 |
| MuSiQue | 8.82 | 8.48 | 6.58 | 7.72 | 7.94 | 14.19 | 12.12 | 11.99 | 13.67 | 12.97 |
| DuReader | 24.06 | 18.30 | 19.11 | 22.64 | 21.05 | 27.34 | 20.03 | 21.07 | 24.07 | 20.78 |
| MultiFieldQA-en | 35.72 | 27.02 | 24.74 | 31.55 | 25.34 | 36.63 | 27.03 | 24.56 | 31.73 | 26.09 |
| MultiFieldQA-zh | 33.32 | 23.84 | 20.13 | 25.69 | 22.55 | 34.13 | 24.90 | 23.11 | 27.49 | 25.44 |
| NarrativeQA | 16.78 | 15.97 | 13.46 | 16.27 | 15.18 | 19.38 | 17.50 | 14.78 | 19.30 | 17.95 |
| Qasper | 17.33 | 15.79 | 15.90 | 16.55 | 16.10 | 26.84 | 23.02 | 21.44 | 24.51 | 21.05 |
| GovReport | 26.25 | 22.54 | 20.73 | 22.84 | 20.47 | 26.21 | 23.89 | 21.78 | 23.76 | 21.54 |
| QMSum | 20.89 | 19.72 | 19.30 | 20.34 | 19.58 | 20.12 | 19.03 | 18.76 | 19.61 | 19.16 |
| MultiNews | 25.83 | 25.07 | 23.32 | 25.16 | 23.27 | 26.06 | 25.38 | 23.71 | 25.33 | 23.79 |
| VCSUM | 14.33 | 13.12 | 12.31 | 13.16 | 12.18 | 16.89 | 15.53 | 14.37 | 16.03 | 13.87 |
| TriviaQA | 83.01 | 83.13 | 80.31 | 83.28 | 81.09 | 88.45 | 88.25 | 85.78 | 88.56 | 85.95 |
| SAMSum | 41.28 | 39.97 | 38.46 | 39.47 | 38.86 | 36.77 | 36.50 | 36.01 | 36.99 | 35.85 |
| TREC | 64.50 | 62.50 | 59.00 | 65.00 | 58.00 | 68.5 | 65.50 | 62.00 | 66.50 | 60.00 |
| LSHT | 17.75 | 17.25 | 15.00 | 16.25 | 15.75 | 20.25 | 20.00 | 18.75 | 19.00 | 18.75 |
| PassageRetrieval-en | 11.50 | 6.50 | 6.50 | 5.00 | 6.50 | 9.00 | 8.50 | 7.50 | 10.00 | 7.00 |
| PassageCount | 4.50 | 5.00 | 5.00 | 5.50 | 5.00 | 3.92 | 3.50 | 2.00 | 2.50 | 3.50 |
| PassageRetrieval-zh | 12.00 | 7.50 | 7.00 | 7.60 | 5.50 | 14.50 | 13.00 | 8.00 | 10.50 | 9.00 |
| LCC | 47.74 | 47.97 | 47.18 | 47.97 | 45.59 | 39.31 | 39.22 | 39.00 | 40.12 | 39.09 |
| RepoBench-P | 44.35 | 43.44 | 43.82 | 43.41 | 43.54 | 42.98 | 41.14 | 39.46 | 41.70 | 38.33 |
| Average | 29.08 | 26.56 | 25.41 | **27.34** | **25.68** | 30.69 | 28.30 | 26.82 | **29.22** | **27.04** |

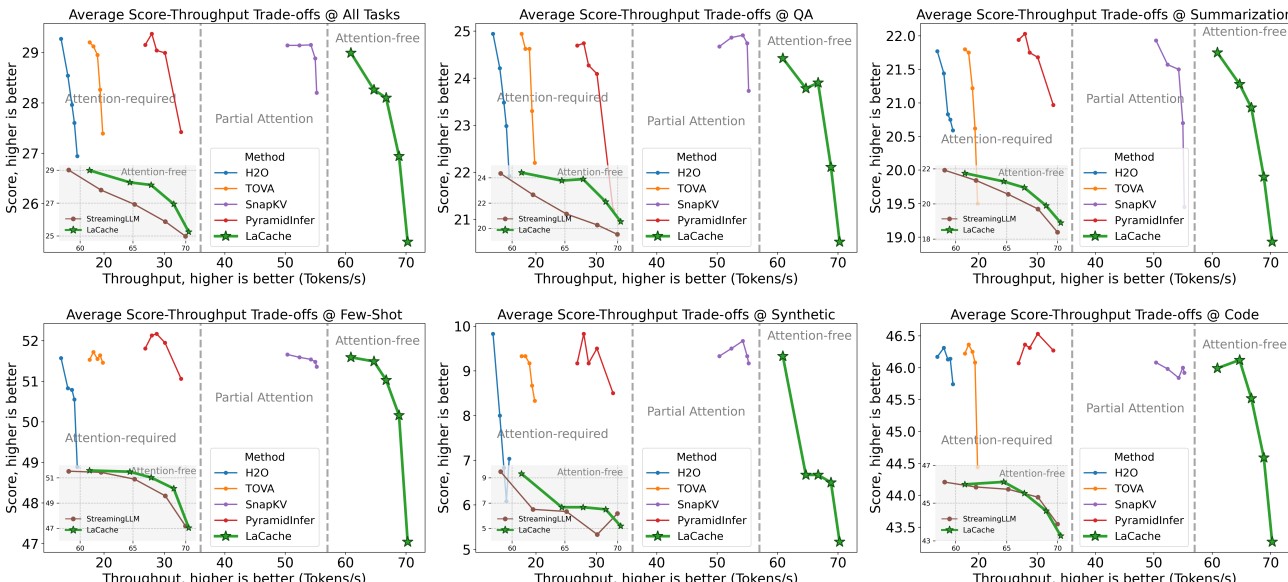

*Figure 7.* Evaluate the score-throughput trade-offs on a single H200 GPU with StreamingLLM (Xiao et al., 2023b), H2O (Zhang et al., 2024), TOVA (Oren et al., 2024), PyramidInfer (Yang et al., 2024), and SnapKV (Li et al., 2024) on LongBench (Bai et al., 2023). The top left subfigure presents the average performance across all 21 tasks, while the remaining subfigures demonstrate the sub-task performance on Question Answering, Summarization, Few-shot Learning, Synthetic Task, and Code Completion, respectively.

mance and throughput on LongBench using the LLaMA-2-7B-Chat model. As shown in Fig. 7, the importance-based KV cache eviction methods H2O (Zhang et al., 2024), TOVA (Oren et al., 2024), PyramidInfer (Yang et al., 2024),

and SnapKV (Li et al., 2024) maintain good scores but suffers from low throughput, while the recency-based method StreamingLLM (Xiao et al., 2023b) experiences lower scores. Our method achieves a better trade-off between task

*Table 4.* Evaluate LaCache on SmolLM2-1.7B-Instruct across 21 LongBench Datasets under 50% and 25% KV cache budgets.

| Cache Budget | 100% | StreamingLLM | | LaCache | |
|---|---|---|---|---|---|
| | | 50% | 25% | 50% | 25% |
| HotpotQA | 24.08 | 24.22 | 22.21 | 24.32 | 22.81 |
| 2WikiMultihopQA | 24.15 | 22.71 | 21.71 | 22.28 | 21.34 |
| MuSiQue | 8.33 | 9.47 | 9.30 | 10.28 | 8.43 |
| DuReader | 20.24 | 14.56 | 14.58 | 16.73 | 15.56 |
| MultiFieldQA-en | 38.78 | 33.39 | 30.28 | 33.35 | 30.09 |
| MultiFieldQA-zh | 16.82 | 14.63 | 12.12 | 13.94 | 12.27 |
| NarrativeQA | 12.65 | 12.62 | 11.68 | 11.98 | 10.73 |
| Qasper | 17.22 | 16.52 | 14.47 | 16.48 | 16.79 |
| GovReport | 26.74 | 21.38 | 18.94 | 21.80 | 18.86 |
| QMSum | 21.86 | 21.37 | 21.04 | 21.40 | 20.87 |
| MultiNews | 25.67 | 25.42 | 24.75 | 25.54 | 24.88 |
| VCSUM | 10.56 | 11.76 | 10.43 | 11.97 | 10.52 |
| TriviaQA | 80.59 | 78.87 | 78.83 | 80.36 | 78.62 |
| SAMSum | 22.05 | 24.15 | 26.12 | 24.63 | 27.80 |
| TREC | 56.00 | 54.50 | 52.50 | 54.50 | 53.00 |
| LSHT | 9.00 | 6.00 | 5.25 | 7.00 | 6.00 |
| PassageRetrieval-en | 9.50 | 5.50 | 5.00 | 6.50 | 5.50 |
| PassageCount | 1.50 | 1.00 | 1.50 | 1.50 | 2.00 |
| PassageRetrieval-zh | 7.77 | 4.86 | 4.92 | 7.42 | 4.17 |
| LCC | 37.76 | 37.85 | 37.75 | 37.99 | 37.83 |
| RepoBench-P | 34.37 | 33.67 | 34.14 | 34.12 | 33.82 |
| Average | 24.07 | 22.59 | 21.79 | **23.05** | **22.00** |

performance and throughput compared to these baselines.

**Extend to small LMs.** To demonstrate LaCache's effectiveness across varying model sizes, we further apply it to the small LM SmolLM2-1.7B-Instruct. As shown in Tab. 4, LaCache consistently achieves higher accuracy than the baseline StreamingLLM under the same cache budgets.

**Benchmark on Needle-In-A-Haystack.** The Needle-In-A-Haystack benchmark (Fu et al., 2024a) assesses LLMs' capabilities of retrieving specific information ("the needle") embedded within extremely long text ("the haystack"), which is crucial for applications requiring precise information retrieval from long context. We benchmark LaCache and StreamingLLM on Llama3.2-3B-Instruct-128k and LongChat-7b-v1.5-32k as shown in Fig. 8 and Fig. 9.

The results demonstrate that LaCache nearly doubles the test accuracy compared to StreamingLLM under the same cache budget—for example, from 54.54% to 99.16% on the Llama3.2-3B-Instruct-128k model under a 50% cache budget and from 33.40% to 65.30% on the LongChat-7b-v1.5-32k model under a 25% cache budget.

**Benchmark on RULER.** The RULER (Hsieh et al., 2024) benchmark utilizes synthetic examples to evaluate long-context LLMs. It encompasses four task categories, including retrieval, multi-hop tracing, aggregation, and question answering. We benchmark LaCache and StreamingLLM on the LongChat-7b-v1.5-32k model under a 50% cache setting, as shown in Tab. 5.

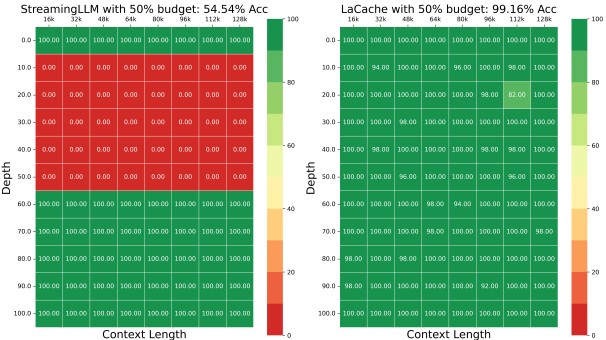

*Figure 8.* Benchmark LaCache and StreamingLLM on Needle-In-A-Haystack (Fu et al., 2024a) using Llama3.2-3B-Instruct-128k (Dubey et al., 2024) under a 50% cache budget setting. Greener indicates better performance.

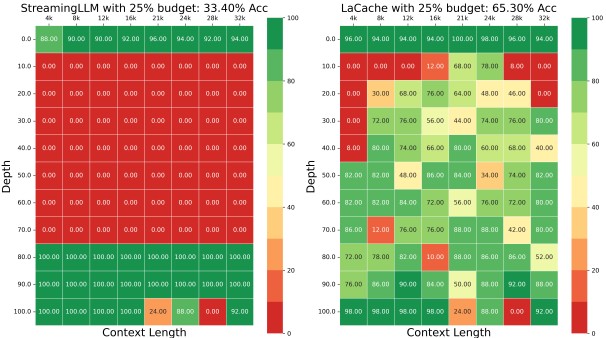

*Figure 9.* Benchmark LaCache and StreamingLLM on Needle-In-A-Haystack (Fu et al., 2024a) using LongChat-7b-v1.5-32k (Li et al., 2023) under a 25% cache budget setting. Greener indicates better performance.

The experimental results further validate the consistently better performance of LaCache under the same KV cache conditions. Specifically, LaCache achieves a 5.06% higher average accuracy across 13 different tasks, with particularly large improvements on the *vt* and *cwe* tasks, where it significantly outperforms the baseline.

### 4.4. Ablation Studies

**Hyperparameter *Span S*.** In long-context understanding tasks such as LongBench, *Span S* is set as an integer approximately equal to the number of layers multiplied by the overall compression ratio, aiming for a uniform compression ratio distribution. For example, under a 50% cache budget, setting $S$ equal to half the number of model layers results in a ∼50% compression ratio across different positions, avoiding situations where some locations are over-compressed while others are under-compressed. In language modeling tasks, $S$ is set to 1/4 of the number of model layers, which was given by the empirical results from our ablation studies, as shown in Fig. 10, where Llama2-7B-Chat model under a 256 KV cache budget and the Wikitext-2 dataset are used

*Table 5.* Evaluate LongChat-7b-v1.5-32k model (Li et al., 2023) on the RULER benchmark (Fu et al., 2024a) under a 50% cache budget setting. A higher number indicates better performance.

| Task | single₁ | single₂ | single₃ | multikey₁ | multikey₂ | multikey₃ | multivalue | multiquery | vt | cwe | fwe | qa₁ | qa₂ | **Avg.** |
|---|---|---|---|---|---|---|---|---|---|---|---|---|---|---|
| StreamingLLM | 45.00 | 49.00 | 45.00 | 53.00 | 50.00 | 45.00 | 47.00 | 42.00 | 29.40 | 17.20 | 42.00 | 75.00 | 43.00 | 44.82 |
| LaCache | 57.00 | 43.00 | 26.00 | 52.00 | 64.00 | 31.00 | 62.75 | 50.25 | 60.80 | 61.00 | 45.67 | 67.00 | 41.00 | **50.88** |

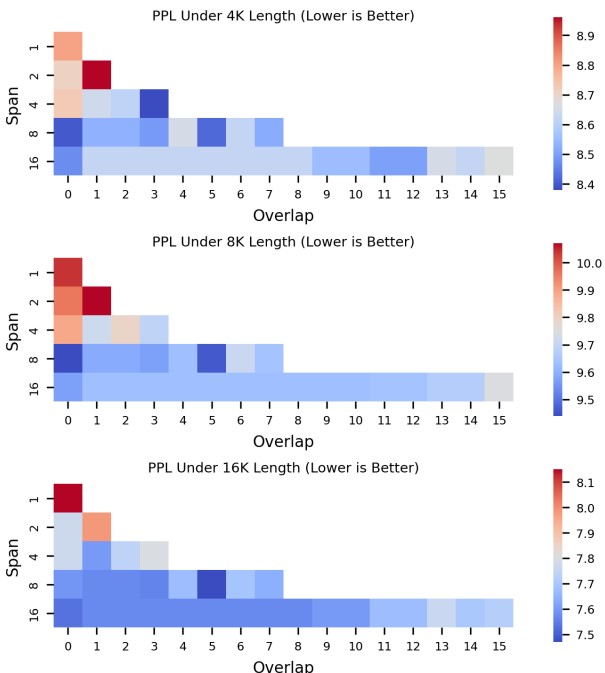

*Figure 10.* Ablation studies on the hyperparameters for language modeling. Perplexity (lower is better) is reported in the figure.

*Table 6.* Ablation studies on the hyperparameters for long-context understanding. Scores (higher is better) are reported in the table.

| Setting | O = 0 | O = S/4 | O = S/2 | Δ(S/4 - 0) | Δ(S/2 - 0) |
|---|---|---|---|---|---|
| QA tasks | 19.48 | 18.94 | 18.48 | -0.54 | -1.00 |
| Synthetic tasks | 5.17 | 5.67 | 6.17 | +0.50 | +1.00 |

for examining the impact of hyperparameters.

**Hyperparameter *Overlap O*.** The choice of *Overlap O* depends on the task type. Specifically, a larger $O$ allows the information of a single token to be distributed across more positions, which is better suited for tasks requiring complex semantic understanding and greater global context. In contrast, a small overlap concentrates the information in fewer positions, which is more appropriate for tasks where the answers appear in a very narrow window. For language modeling tasks, $O$ is set to 1/2 of $S$ for achieving better semantic continuity. For long-context understanding tasks, as shown in Tab. 6, a larger overlap consistently improves performance on tasks that require more global information, such as synthetic tasks (PassageCount, PassageRetrieval-en, and PassageRetrieval-zh) while reducing performance on tasks

that rely more on local information, such as QA tasks (NarrativeQA, Qasper, MultiFieldQA-en, and MultiFieldQA-zh).

## 5. Limitations and Future Work

While LaCache demonstrates advantages for accurate and efficient long-context generation in LLMs, it also has limitations that present opportunities for future exploration: Although the ladder-shaped KV cache pattern is effective, it may not be optimal for every scenario. Alternative patterns could further enhance memory efficiency and performance. Future work can explore diverse KV storage configurations based on our core insight: recent tokens are crucial for generation accuracy, but fewer layers might suffice for effectively processing and storing their KV caches. Additionally, LaCache is implemented in a training-free setting to ensure efficiency and ease of deployment. Incorporating fine-tuning could further improve performance by adapting the KV cache pattern to specific tasks. Future work can extend LaCache to support fine-tuning and benchmark its performance against training-dependent methods.

## 6. Conclusion

In this work, we introduce LaCache, a novel, training-free, and easy-to-deploy KV cache optimization framework designed to enhance both the efficiency and effectiveness of LLMs in long-context generation tasks. LaCache addresses the limitations of existing KV caching methods through a ladder-shaped KV cache storage pattern and an iterative compaction mechanism. These innovations enable LLMs to better capture long-range dependencies, optimize memory usage, and sustain continuous generation even under fixed storage constraints. Specifically, by sequentially storing KV pairs both within and across layers, the ladder-shaped structure allows the model to preserve crucial information at different levels of context. Additionally, the iterative compaction mechanism dynamically manages memory, ensuring that essential information is prioritized. Our results show that LaCache significantly improves memory efficiency while maintaining high generation quality, outperforming baseline methods across various benchmarks. By offering a scalable and storage-efficient solution, LaCache enhances the capacity of LLMs to manage extended contexts and continuous generation, paving the way for further innovations in long-range LLM optimization.

## Acknowledgment

This work was partially supported by the National Science Foundation (NSF) through the Computing and Communication Foundations (CCF) program (Award ID: 2400511), the Division of Information & Intelligent Systems (IIS) program (Award ID: 2403297), and CoCoSys, one of the seven centers in JUMP 2.0, a Semiconductor Research Corporation (SRC) program sponsored by DARPA. It was also supported by the Department of Health and Human Services Advanced Research Projects Agency for Health (ARPA-H) under Agreement Number 140D042490003. The views and conclusions contained herein are those of the authors and should not be interpreted as necessarily representing the official policies or endorsements, either expressed or implied, of the Advanced Research Projects Agency for Health or the U.S. Government.

## Impact Statement

This work aims to enhance the generation efficiency of LLMs on long-context tasks, achieving continuous generation without OOM while maintaining long-context accuracy. As such, it can facilitate wider use of LLMs and does not suffer from additional societal consequences if LLMs are properly used.

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
