# OpenReview forum: "LaCache: Ladder-Shaped KV Caching for Efficient Long-Context Modeling of Large Language Models"
_ICML.cc/2025/Conference — ICML 2025 poster_

### Official Review · Reviewer_gKML · 2025-03-10

**Overall Recommendation:** 1

**Summary:**

the paper presented a training-free KV cache optimization, named LaCache, for long-text generation tasks.  the proposed framework employs ladder-shaped KV cache storage pattern and an iterative compaction mechanism to enable LLMs to better capture long-range dependencies, optimize memory usage, and sustain continuous generation. the authors evaluated the proposed framework by some experiments.

**Claims And Evidence:**

the experimental results show that the proposed framework is effective. However, the improvement is not significant.

**Essential References Not Discussed:**

no

**Experimental Designs Or Analyses:**

the authors should evaluate the proposed method on more benchmarks, such as Needle-in-a-Haystack, and RULER.and on recent baselines.

**Methods And Evaluation Criteria:**

More datasets should to be used in the experiments. For example, Needle-in-a-Haystack, and RULER benchmarks.

**Other Comments Or Suggestions:**

No

**Other Strengths And Weaknesses:**

the main weakness of the paper is that it lack of theoretical  analysis and discussion.

**Questions For Authors:**

1. the rationale of Ladder-Shaped KV Cache Pattern is not convincing.  Theoretical analysis and discussion should be provided.  I don't fully understand why the authors presented the two formulas in page 4. The formulas don't provide convincing theoretical support for the proposed method.
2. What is the KV cache budget for LaCache in Figure 7?
3. As shown in Figure 7，LaCache is not superior to  TOVA. TOVA have remarkable advantage on  F1 score while its throughtput is about 17.5, which is not much smaller than 30 (that of LaCache).

**Relation To Broader Scientific Literature:**

incremental work

**Theoretical Claims:**

No theoretical claims is shown in this paper.

---

> ### Author Rebuttal · Authors · 2025-04-01
>
> Thank you for your time and constructive suggestions! We have addressed all your comments and suggestions as follows.
>
> ---
>
> **Q1. Evaluation on more datasets: Needle-in-a-Haystack (NIAH) & RULER**
>
> Following your suggestions, we have added experiments on the NIAH and RULER datasets. Please check our [response to Reviewer fB6E's Question 1](https://openreview.net/forum?id=SDjZtxDo35&noteId=jrewAScmqH).
>
> ---
>
> **Q2. Add more KV cache compression baselines**
>
> Following your suggestions, we have added comparisons with SnapKV[1] and PyramidInfer[2] on the entire LongBench dataset, as shown in [this figure https://ibb.co/q3GrqCZQ](https://ibb.co/q3GrqCZQ). This set of results consistently validates that LaCache achieves better score-throughput trade-offs than various baselines.
>
> ---
>
> **Q3. Comparison with TOVA in Figure 7**
>
> First, we humbly clarify and emphasize that one highlight of LaCache is its simplicity and seamless integration with the existing FlashAttention implementation, as validated by our method’s improved accuracy over baselines under the same real-device efficiency.
>
> Following your suggestion, we conducted experiments comparing with TOVA under more various cache budgets in [figure https://ibb.co/q3GrqCZQ](https://ibb.co/q3GrqCZQ). These experiments demonstrate that LaCache achieves better average scores than attention-based methods by a large margin under the same real-device throughput.
>
> ---
>
> **Q4. The KV cache budget for LaCache in Figure 7**
>
> We benchmarked our method against different baselines under budgets ranging from 20% to 50% in the original Figure 7. Additionally, our [updated Figure 7](https://ibb.co/q3GrqCZQ) includes more baselines and a wider range from 20% to 99%.
>
> ---
>
> **Q5. The rationale and theoretical analyses of the ladder-shaped KV cache pattern**
>
> We humbly clarify that the analysis on Page 4 of our manuscript is intended to provide a high-level rationale for why LaCache is effective rather than serving as theoretical proof. This analysis demonstrates that the ladder pattern can cover potentially important tokens more effectively under the same token budget, thus enhancing the lower bound of information retention, particularly when compared to assigning the same KV tokens across all layers.
>
> This analysis has been appreciated by @Reviewer cTxa, who recognized it as offering “good insights” and showing “a deeper understanding of the problem.” Additionally, as noted by @Reviewer fB6E - “I particularly like the analysis described by Figure 3, where the caching pattern is compared to randomly chosen caching patterns across cache sizes”, we have empirically verified that the proposed KV cache is robust and close to optimal. Specifically, in Figure 3 of our manuscript, we randomly generated over 1,500 patterns under different KV cache sizes to explore all possible configurations where the proposed ladder pattern lies on the Pareto optimality boundary.
>
> In summary, this work primarily aims to (1) empirically demonstrate that our method lies on the Pareto frontier, consistent with the evaluation protocols of most KV cache compression methods, and (2) provide the generalizable insight that the same KV token does not need to be maintained across all layers. A more rigorous theoretical proof of our method is planned for future work.
>
> ---
>
> **Q6. Regarding “incremental”**
>
> We humbly clarify that, given the increasing need for LLMs’ long-context capability in more real-world AI serving systems and applications, simplicity is a key highlight and design consideration of our proposed method. In particular, our proposed technique can (1) be seamlessly compatible with FlashAttention without introducing extra overhead during inference and (2) offer generalizable insights to guide future KV cache designs—i.e., the same KV token does not need to be maintained in all layers, and the ladder pattern is an effective way to achieve this by improving the lower bound of information retention to cover potentially important tokens. This has been positively recognized by @Reviewer cTxa, who noted, “The effort for building LaCache’s practical considerations is very helpful as it could be deployed easily with existing systems. Great to see the compatibility with systems such as Flash Attention. Very promising work.”
>
> Finally, we wish to emphasize that developing a simple yet effective technique like LaCache is nontrivial, given the extensive research already conducted in this area. This sentiment is echoed by @Reviewer fB6E, who noted, “This is a crowded space, but I think this is a nice contribution and argues its point well.”
>
> ---
>
> Thank you for your thoughtful suggestions and comments that aim to help strengthen this paper! If you have any further questions or updated comments, we would really appreciate and be happy to address them.
>
> ---
>
> **References**
>
> [1] SnapKV: LLM Knows What You are Looking for Before Generation
>
> [2] PyramidInfer: Pyramid KV Cache Compression for High-throughput LLM Inference

---

### Official Review · Reviewer_QVQG · 2025-03-11

**Overall Recommendation:** 2

**Summary:**

This paper proposes a method to compress KV cache with the goal of storing different sets of tokens in different layers, termed Ladder-Shaped KV Cache (LaCache). The idea is to keep earlier tokens in the sequence in the lower layer and the later tokens in the deeper layer, which intuitively makes sense.

**Claims And Evidence:**

Experiments are conducted on the language modeling task (Wikitext and PG19) and tasks from LongBench for Llama-2-7B and Llama-3-8B. Results show that LaCache performs better than StreamingLLM on both tasks. However, I think the experiment is a little light and can be strengthened with more recent long-context models (see section on "Experimental Designs Or Analyses") and more baselines (see section on "Essential References Not Discussed").

**Essential References Not Discussed:**

Missing citation and related work:
* [SnapKV](https://arxiv.org/abs/2404.14469) (NeurIIPS 2024) is a relevant method which compresses KV cache based on input tokens' attention pattern, which is a stronger and more up-to-date baseline compared to H2O and StreamingLLM.
* [PyradmicInfer](https://aclanthology.org/2024.findings-acl.195.pdf) (ACL 2024) also proposes a layer specific KV cache eviction-based method and is a relevant baseline.

**Experimental Designs Or Analyses:**

The paper primarily evaluated short context models (Llama-2-7B and Llama-3.1-8B) with up to 8K context length. These models inherently do not perform well on LongBench as most tasks have input exceeding the context length (e.g. for the PassageRetrieval task). As the proposed method aims to compress KV cache, it will be more suitable for evaluate on long-context models that can take longer input (e.g. Llama-3.1 and Qwen-2 models).

**Methods And Evaluation Criteria:**

The benchmark datasets (PG19 and Longbench) make sense.

**Other Comments Or Suggestions:**

* Clarity of the proposed method: While figure 2 and figure 4 illustrate the procedure for LaCache, it would be helpful to include a pseudocode or algorithm to formally describe how the algorithm works.

**Other Strengths And Weaknesses:**

N/A

**Questions For Authors:**

* While it is not mentioned in the paper, I assume the compression happens at decoding time -- i.e. the model encodes the entire input for LongBench and perform LaCache compression. Is the understanding correct?
* What are the $S$ and $O$ used in the experiments of language modeling and LongBench experiments and how are they decided?

**Relation To Broader Scientific Literature:**

This work contributes to the line of research on memory efficiency of long-context inference through KV cache compression, which is an active research area.

**Theoretical Claims:**

N/A

---

> ### Author Rebuttal · Authors · 2025-04-01
>
> Thank you for your time and constructive suggestions! We have addressed all your comments and suggestions as follows.
>
> ---
>
> **Q1. Experiments on longer-context models trained on longer inputs**
>
> Thank you for the suggestion! We have added experiments using both Llama3.2-3B-Instruct-128k and LongChat-7b-v1.5-32k to verify the consistent effectiveness of LaCache.
>
> On the Needle In A Haystack (NIAH) benchmark, we evaluated both Llama3.2-3B-Instruct-128k (in [figure https://ibb.co/1GjhX0b5](https://ibb.co/1GjhX0b5)) and LongChat-7b-v1.5-32k (in [figure https://ibb.co/s9qp1s2K](https://ibb.co/s9qp1s2K)) under 50% and 25% cache budget settings. Our results demonstrate that LaCache nearly doubles the test accuracy compared to StreamingLLM under the same cache budget—for example, from 54.54% to 99.16% on the Llama3.2-3B-Instruct-128k model under a 50% cache budget, and from 33.40% to 65.30% on the LongChat-7b-v1.5-32k model under a 25% cache budget.
>
> ---
>
> **Q2. Add more KV cache compression baselines**
>
> Following your suggestions, we have added comparisons with both SnapKV [1] and PyramidInfer [2] as shown in [figure https://ibb.co/q3GrqCZQ](https://ibb.co/q3GrqCZQ). This set of results consistently validates that LaCache achieves better score-throughput trade-offs across various tasks on the LongBench benchmark.
>
> ---
>
> **Q3. LaCache implementation and pseudocode**
>
> Thank you for the constructive suggestion! We will include a more comprehensive version along with additional implementation details in our final manuscript.
>
> ---
>
> **Q4. When LaCache is applied**
>
> Yes, you are correct -  it is applied at every step of decoding. Specifically, after prefilling the entire input, LaCache is used to reduce the KV cache and maintain it at a constant size by applying LaCache after each decoding step.
>
> ---
>
> **Q5. Clarifications on the hyperparameters S and O**
>
> The definitions of S and O are introduced at the end of Section 3.2 in our submitted manuscript.
>
> (a) In long-context understanding tasks such as LongBench, S is set as an integer approximately equal to the number of layers multiplied by the overall compression ratio, aiming for a uniform compression ratio distribution. For example, under a 50% cache budget, setting the span equal to half the number of model layers results in a ~50% compression ratio across different positions, helping to avoid situations where some locations are over-compressed while others are under-compressed. In language modeling tasks, S is set to 1/4 of the number of model layers, which was given by the empirical results from our ablation studies, as shown in Figure 8.
>
> (b) The choice of O depends on the task type. Specifically, a larger overlap (O) allows the information of a single token to be distributed across more positions, which is better suited for tasks requiring complex semantic understanding and greater global context. In contrast, a small overlap concentrates the information in fewer positions, which is more appropriate for tasks where the answers appear in a very narrow window.
>
> |                 | Overlap=0 | Overlap=Span/4 | Overlap=Span/2 | $\Delta$ (Span/4 - 0) | $\Delta$ (Span/2 - 0) |
> |-----------------|------------|------------------|------------------|----------------------|----------------------|
> | QA tasks        | 19.48      | 18.94            | 18.48            | -0.54                | -1.00                |
> | Synthetic tasks | 5.17       | 5.67             | 6.17             | +0.50                 | +1.00                 |
>
>
> For language modeling tasks, S is set to 1/2 of Span because of better semantic continuity. For long-context understanding tasks, following your suggestion, we have added above experiments on LongBench to demonstrate the impact of the overlap parameter O. As shown in the table, a larger overlap consistently improves performance on tasks that require more global information, such as synthetic tasks (PassageCount, PassageRetrieval-en, and PassageRetrieval-zh), while reducing performance on tasks that rely more on local information, such as QA tasks (NarrativeQA, Qasper, MultiFieldQA-en, and MultiFieldQA-zh).
>
> ---
>
> **Q6. Missing citations and related work**
>
> Thank you for pointing this out! We have added experiments on these two baselines as shown in [figure https://ibb.co/q3GrqCZQ](https://ibb.co/q3GrqCZQ) and will include these two related works in our final manuscript.
>
> ---
>
> Thank you for your thoughtful suggestions that aim to help strengthen this paper! If you have any further questions or updated comments, we would really appreciate it and be happy to address them.
>
> ---
>
> **References**
>
> [1] SnapKV: LLM Knows What You are Looking for Before Generation
>
> [2] PyramidInfer: Pyramid KV Cache Compression for High-throughput LLM Inference

---

### Official Review · Reviewer_cTxa · 2025-03-14

**Overall Recommendation:** 4

**Summary:**

* Paper proposed a training-free KV Cache compression which stores KV pairs not only sequentially (left-to-right within
each layer) but also across layers (from shallow to deep), giving deeper capabilities to capture long-range dependencies.
*  Proposes iterative compaction mechanism that progressively compresses older caches, freeing up space for new tokens within a fixed cache size.

**Claims And Evidence:**

* Different layers can maintain the KV Cache corresponding to different sets of tokens. This is a good insight.
* The ladder-shaped method is both intuitive to follow and well-developed.
* Iterative method to remove old KV Cache states to ensure memory compaction is a good idea.
* Figure 4 is useful visualization of the compaction technqiue.

**Essential References Not Discussed:**

All references are discussed and most relevant works have been used appropriately as baselines.

**Experimental Designs Or Analyses:**

* "Continuously extending a repetitive pattern and assigning coverage to each layer as equally as possible, which is ensured by our ladder-shaped pattern, improves the lower bound of the above minimax optimality than an unequal coverage strategy." This is a good insight.
* "An ever-expanding ladder pattern with partial overlaps creates a smoother fade-out of older tokens, helping to maintain stable information retention", also shows a deeper understanding of the problem.
* The ablation studies section is a little weak, considering only two hyperparameters. It is not clear yet if the parameters are both necessary and complete?

**Methods And Evaluation Criteria:**

* Experiments are carried out on standard long context benchmarks  (LongBench) and LaCache shows very promising results.
* Comparison against both training-based methods (StreamingLLM) and training-free methods (H20, TOVA) is very helpful.
* Evaluation has been carried out on timely set of models

**Other Comments Or Suggestions:**

N/A

**Other Strengths And Weaknesses:**

* The effort for building LaCache's practical considerations is very helpful as it could be deployed easily with existing systems. Great to see the compatibility with systems such as Flash Attention. Very promising work.

**Questions For Authors:**

Minor concern: It is not clear if span and overlap are the only parameters which should be considered in the ladder-based design. Would be useful to see some additional analysis on that. Overall, this paper was a very interesting read.

**Relation To Broader Scientific Literature:**

* Both contributions presented are extremely relevant for practical LLM inference systems, the ladder-based compression and the memory compaction, both seem extremely practical.

**Theoretical Claims:**

None

---

> ### Author Rebuttal · Authors · 2025-04-01
>
> Thank you for recognizing the insightfulness and promising results of our work, as well as for the constructive suggestions! We have addressed all your comments and suggestions as follows.
>
> ---
>
> **Q1. More ablation studies and analyses on hyperparameters**
>
> Thank you for the suggestion! We highlight that the ladder pattern can be fully determined by the two hyperparameters: Span and Overlap (i.e., S and O in Section 3.2 of our manuscript).
>
> (a) In long-context understanding tasks such as LongBench, S is set as an integer approximately equal to the number of layers multiplied by the overall compression ratio, aiming for a uniform compression ratio distribution. For example, under a 50% cache budget, setting the span equal to half the number of model layers results in a ~50% compression ratio across different positions, helping to avoid situations where some locations are over-compressed while others are under-compressed. In language modeling tasks, S is set to 1/4 of the number of model layers, which was given by the empirical results from our ablation studies, as shown in Figure 8.
>
> (b) The choice of O depends on the task type. Specifically, a larger overlap (O) allows the information of a single token to be distributed across more positions, which is better suited for tasks requiring complex semantic understanding and greater global context. In contrast, a small overlap concentrates the information in fewer positions, which is more appropriate for tasks where the answers appear in a very narrow window.
>
> |                 | Overlap=0 | Overlap=Span/4 | Overlap=Span/2 | $\Delta$ (Span/4 - 0) | $\Delta$ (Span/2 - 0) |
> |-----------------|------------|------------------|------------------|----------------------|----------------------|
> | QA tasks        | 19.48      | 18.94            | 18.48            | -0.54                | -1.00                |
> | Synthetic tasks | 5.17       | 5.67             | 6.17             | +0.50                 | +1.00                 |
>
>
> For language modeling tasks, S is set to 1/2 of Span because of better semantic continuity. For long-context understanding tasks, following your suggestion, we have added above experiments on LongBench to demonstrate the impact of the overlap parameter O. As shown in the table, a larger overlap consistently improves performance on tasks that require more global information, such as synthetic tasks (PassageCount, PassageRetrieval-en, and PassageRetrieval-zh), while reducing performance on tasks that rely more on local information, such as QA tasks (NarrativeQA, Qasper, MultiFieldQA-en, and MultiFieldQA-zh).
>
>
> ---
>
> Thank you for your thoughtful suggestions to help strengthen this paper! If you have any further questions or updated comments, we would really appreciate it and  be happy to address them.

---

### Official Review · Reviewer_fB6E · 2025-03-14

**Overall Recommendation:** 4

**Summary:**

The paper introduces LaCache, a scheme for progressive cache eviction for more efficient long context processing. Rather than evicting the same tokens at each layer, LaCache evicts tokens using a ladder-like scheme so that earlier layers maintain tokens from earlier in the context and later layers retain tokens from later in the context (with the immediate local context being fully preserved). They show that this scheme reduces the performance degradation from cache eviction and is more efficient than attention-based patterns for cache eviction; the size of each ladder "rung" and the overlap between rungs across layers are hyperparameters, which are analyzed in the analysis.

**Claims And Evidence:**

Yes; the paper demonstrates the strong performance relative to reasonable baselines in terms of both downstream metrics and latency, compared with a fixed storage (KV cache size) budget.

**Essential References Not Discussed:**

Not that I am aware of.

**Experimental Designs Or Analyses:**

Yes; I think the experimental design is reasonable and compared against good baselines.

I do think it would be nice to see results on the baselines other than StreamingLLM in the main table, especially as it seems that H2O might sometimes outperform StreamingLLM in downstream performance if not in latency.

**Methods And Evaluation Criteria:**

Yes; I think the use of perplexity alone would not be convincing, so I appreciate the use of a diverse set of tasks from LongBench. It would be nice to show a non-perplexity task in the very long (>>16k) context regime; however, I understand there are not all that many tasks that fit this description.

**Other Comments Or Suggestions:**

N/A

**Other Strengths And Weaknesses:**

I particularly like the analysis described by Figure 3, where the caching pattern is compared to randomly chosen caching patterns across cache sizes. I think this is a nice extra that makes a strong argument for this technique.

**Questions For Authors:**

Q1. Can you report how LaCache compares to H2O on the remainder of the LongBench tasks you use?

**Relation To Broader Scientific Literature:**

This is a crowded space, but I think this is a nice contribution and argues its point well. This is a type of fixed pattern I haven't seen before-- most methods evict the same tokens across layers, or use an attention-based decision at each layer. Using earlier layers for earlier tokens is an interesting idea.

**Theoretical Claims:**

Not exactly a proof, more an informal optimization argument-- but I'm not fully convinced by the argument about the costs of compression patterns. In particular, since we are not working with tokens, but with contextual embeddings, it seems that some information about the "important tokens" will be present even if the embeddings of those tokens are not chosen; in addition, I'm not convinced that you can choose an ideal set of "important tokens" to maximize performance in the first place. This is not central to the paper, so it wasn't a major factor in my score; however, I don't think this section adds much to the argument.

---

> ### Author Rebuttal · Authors · 2025-04-01
>
> Thank you for recognizing the contributions and analysis offered by our work, as well as for the constructive suggestions! We have addressed all your comments and suggestions as follows.
>
> ---
>
> **Q1. Experiments on non-perplexity tasks with a long context regime (>>16k)**
>
> Following your suggestions, we have added new experiments with context lengths up to 128k on both Llama3.2-3B-Instruct-128k and LongChat-7b-v1.5-32k to verify the consistent effectiveness of LaCache.
>
> (a) On the Needle In A Haystack (NIAH) benchmark, we evaluated both Llama3.2-3B-Instruct-128k (in [figure https://ibb.co/1GjhX0b5](https://ibb.co/1GjhX0b5)) and LongChat-7b-v1.5-32k (in [figure https://ibb.co/s9qp1s2K](https://ibb.co/s9qp1s2K)) under both 50% and 25% cache budget settings. Our results demonstrate that LaCache nearly doubles the test accuracy compared to StreamingLLM under the same cache budget—for example, from 54.54% to 99.16% on the Llama3.2-3B-Instruct-128k model under a 50% cache budget, and from 33.40% to 65.30% on the LongChat-7b-v1.5-32k model under a 25% cache budget.
>
> | Task               | StreamingLLM | LaCache |
> |--------------------|--------------|---------|
> | niah_single_1      | 45.0         | 57.0    |
> | niah_single_2      | 49.0         | 43.0    |
> | niah_single_3      | 45.0         | 26.0    |
> | niah_multikey_1    | 53.0         | 52.0    |
> | niah_multikey_2    | 50.0         | 64.0    |
> | niah_multikey_3    | 45.0         | 31.0    |
> | niah_multivalue    | 47.0         | 62.75   |
> | niah_multiquery    | 42.0         | 50.25   |
> | vt                 | 29.4         | 60.8    |
> | cwe                | 17.2         | 61.0    |
> | fwe                | 42.0         | 45.67   |
> | qa_1               | 75.0         | 67.0    |
> | qa_2               | 43.0         | 41.0    |
> | **Mean**           | **44.82**    | **50.88** |
>
>
> (b) Similarly, on the RULER benchmark, we evaluated the LongChat-7b-v1.5-32k model under a 50% cache setting. The experimental results above verify the advantageous performance of LaCache under the same KV cache. Specifically, LaCache achieves a 5.06% higher average accuracy across 13 different tasks, especially in task cwe and fwe, where LaCache outperforms the baseline by a large margin.
>
>
> ---
>
> **Q2. More analysis on the costs of compression patterns**
>
> Thank you for your insightful comments! We agree that selecting an ideal set of "important tokens" to maximize performance is not the primary goal of our method.
>
> The key point we aim to convey in this analysis is that LaCache improves over baselines by leveraging the insight that it is not necessary to maintain the same set of KV tokens across all layers. In particular, LaCache alleviates this redundancy through the proposed ladder pattern, which can more effectively cover potentially important tokens within a given budget and thus improve the lower bound of information retention. We will incorporate your suggestion and clarify this more clearly in the final version.
>
> ---
>
> **Q3. Add more baselines on LongBench**
>
> Thank you for your suggestion! We have added benchmarks with recent baselines, including both SnapKV [1] and PyramidInfer [2], as shown in [figure https://ibb.co/q3GrqCZQ](https://ibb.co/q3GrqCZQ). This set of results consistently validates that our LaCache achieves better score-throughput trade-offs across various tasks on the LongBench benchmark.
>
>
> ---
>
> **Q4. Benchmark LaCache with H2O on the remainder of the LongBench tasks**
>
> Following your suggestion, we have updated the overall performance comparison between LaCache and H2O on the entire 21 tasks of the LongBench benchmark as shown in [figure https://ibb.co/q3GrqCZQ](https://ibb.co/q3GrqCZQ), which demonstrates that LaCache achieves better F1-score-throughput trade-offs across various tasks on the full LongBench benchmark. We will also report results by categories in the final version.
>
> ---
>
> Thank you for your thoughtful suggestions to help strengthen this paper! If you have any further questions or updated comments, we would really appreciate it and be happy to address them.
>
> ---
>
> **References**
>
> [1] SnapKV: LLM Knows What You are Looking for Before Generation
>
> [2] PyramidInfer: Pyramid KV Cache Compression for High-throughput LLM Inference

---

### Decision · Program_Chairs · 2025-05-01

**Decision:**

Accept (poster)

**Comment:**

This paper present LaCache to compress the KV cache with comprehensive empirical validation. The ladder-shaped eviction strategy appeared to be novel. The reviews were quite diverse, with 2 accept and 1 weak reject and 1 reject. I believe the authors did a reasonable good job to address the concerns including both reviewer QVQG and gKML. One remaining concern was about the empirical comparison with SnapKV, whose accuracy appeared to be better than LaCache in the figure with slower speed. I recommend the authors to finish the remaining Pareto figure (as well as have a balanced discussion) to fully address gKML's accuracy concern.